# Utilization of accessible resources in the fabrication of an affordable, portable, high-resolution, 3D printed, digital microscope for Philippine diagnostic applications

**Mark Kristan Espejo Cabello, Jeremie E. De Guzman** *

Research Faculty, Ateneo de Manila University School of Medicine and Public Health, Center for Research and Innovation, Pasig City, National Capital Region, Philippines

* jedeguzman@ateneo.edu

**Data Availability Statement:** The detailed instructions, bill of materials, and the STL files needed to build the Openflexure Microscope (OFM)

## Abstract

Philippine clinical laboratory licensing requirements mandate that diagnostic microscopy for Tuberculosis (TB) sputum microscopy, urinalysis, pap smears, wet smears, an option for complete blood count, stool exams, and malaria thick and thin smears should be accessible and available in health facilities including primary care centers. However, access to these essential diagnostics is hampered by the lack of trained personnel, relatively high costs for supplies and equipment especially in rural and underserved areas. This served as motivation for our team to utilize accessible resources in the form of affordable 3D printers, available CAD software, and components to build our low-cost Openflexure microscope (OFM) prototype. We successfully fabricated our prototype for a total of 310$ with a weight of 525g. We used pathology teaching slides from the Ateneo School of Medicine and Public Health and examined the OFM prototype imaging capabilities. The calculated image resolution was 13% higher compared to an LED light microscope sample captured by a mobile phone at 40x and 15% for 100x. The sampled slide images had adequate clarity with some identifiable cellular features for Rheumatic Heart Disease (RHD), Tuberculosis in soft tissue, and Ascariasis. We were able to correct the color aberrations of the OFM we built and was able to scan images up to 1000x magnification without using oil. Given the features and cost, the OFM prototype can be an attractive and affordable option as an alternative or augmentation to diagnostic microscopy in Philippine primary care. Moreover, it may enable telepathology to support diagnostic microscopy in frontline care.

## Introduction

### Availability and access to diagnostic microscopy in the Philippines

Diagnostic microscopy services in the Philippines are an essential part of primary health care. It is included in the licensing requirements for primary care facilities and standalone clinical laboratories. Diagnostic microscopy is used for the following services: 1) sputum microscopy

can be found at their website openflexure.org and at https://gitlab.com/openflexure. For other specific inquiries on our OFM version, please feel free to contact jedeguzman@ateneo.edu.

**Funding:** This study was funded by the University Research Council of Ateneo de Manila University (URC control number: URC 2022-15). JDG is the principal investigator and project leader where the grant funded the procurement of research supplies, equipment, and personnel salary. The grant requires publication of the research outputs in a peer-reviewed, Scopus indexed journal. The grant funders have no role in study design, data collection, analysis, and manuscript preparation. MKC's salary is funded by the URC grant.

**Competing interests:** We have read the journal's policy and JDG declares the following competing interests. He is an active medical device engineer and entrepreneur. He has engaged in multiple paid speaking engagements, consultancies, and is actively seeking commercialization opportunities. MKC declares no competing interest.

for tuberculosis, 2) urinalysis, 3) pap smear, 4) wet smear for *Trichomonas trichiura*, 5) complete blood count with platelet count [1], 6) stool exam and Kato-Katz technique for schistosomiasis and soil-transmitted helminth endemic areas [2], and 7) thick and thin smears for malaria endemic areas [3]. However, the availability, accessibility, and affordability of these common diagnostic services remains a challenge for many parts of the Philippine healthcare system, especially for geographically isolated and disadvantaged areas [4].

Barriers that affect how these services are accessed and hinder its possible expansion include difficulties in procuring consumables, lack of trained personnel to conduct microscopy, the lack of adequate infrastructure and support to house, power, and maintain microscopy equipment. Another barrier that is important to consider is the cost of acquisition and maintenance of microscopes and related consumables. These barriers negatively impact overall access and sustainability to essential microscopy-based diagnostics [5]. For instance, a basic light microscope that is acceptable for use in clinical diagnostics starts at roughly 1,300 $ US (at 55 ₱ PHP to 1 $ US exchange rate) [6]. This listed price can increase depending on the specifications, brand, and features.

## Rationale for building the OpenFlexure Microscope

One promising option to increase access to diagnostic microscopy is to fabricate lower-cost digital microscopes using open-source designs and 3D printing. While many designs are available, our team chose the OpenFlexure Microscope (OFM) system to fabricate and localize for the Philippine health system [7]. The OFM is an open-source design as shown in Fig 1.

It was chosen because the cost of fabricating a high-optical, research-grade version was quoted at 200 $ US for the needed parts. Furthermore, this design has been used previously for malaria thick and thin smears [8].

The OpenFlexure Microscope project is an open-source hardware project managed by the Bath Open Instrumentation group from the University of Bath. The project aims to make high precision mechanical positioning using 3d printing for use in microscopes, micromanipulation, and others. The project website can be found at www.openflexure.org, where all the current build instructions, stereolithography files, software, and advice can be found to build different versions of the OFM [9].

The features of the OFM design allows for precise stage movement, auto-focusing, partial slide scanning, saving sampled images, with local and distance internet connectivity [8]. These features would potentially enable approaches like telepathology using either real-time or store and forward methods. With these features, primary care teams could work with pathologists and other experts to aid in interpreting patient samples or to have more training opportunities to improve their skills in diagnostic microscopy [10]. As an open-source design, it also gives our team the ability and flexibility to modify, repair, replace, and upgrade the microscope and its different components as needed depending on local context and needs [11].

Utilizing free and open source hardware and 3D printing technologies could provide operational and economic advantages for the Philippines to rapidly capacitate its design and production capacity for locally built health technologies [12]. Lessons from the COVID-19 pandemic show that overdependence on importation for essential technologies produce health security risks [13]. Furthermore, developing local capacity also has downstream effects such as more responsive devices that are fit-for-purpose and spur innovation [14].

This study demonstrated our ability to utilize digital design software, 3D printing, and accessible components in the Philippine market to fabricate our version of the OFM to rapidly build local manufacturing capacity. Our team also examined the initial performance of our OFM prototype by taking sampled images of pathology teaching slides used in the Ateneo

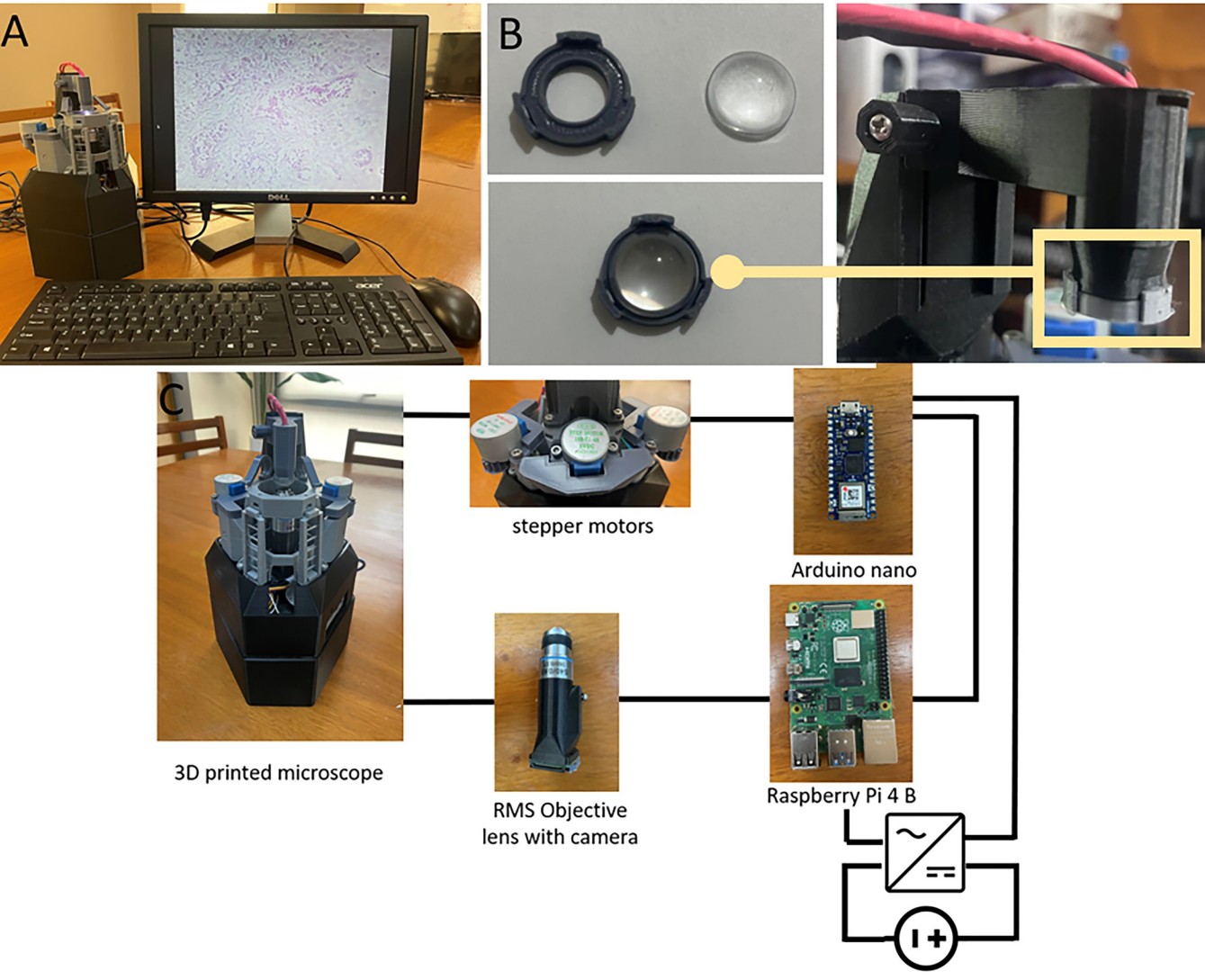

**Fig 1.** A) OFM set-up: 3D printed microscope, display. keyboard and mouse. B) The 3D printed microscope contains modifications such as snap-fit for condenser lens. C) Schematic diagram of the wiring system of the 3D printed microscope.

School of Medicine and Public health. Lastly, we discuss how our version of the OFM could potentially help in improving access and affordability of diagnostic microscopy services in the Philippine healthcare system at the primary care level.

## Materials and methods

Fabrication of the 3D printed microscope consisted of these following steps: 1) printing, 2) assembly, and 3) wiring. Printing methods were based on the works and specifications from the University of Bath OFM project. Detailed build instructions for this microscope can be found in the OFM website at openflexure.org.

Polylactic acid (eSUNs PLA +, 1.75 mm, 1.24 g/cm$^3$, 1000 g) was used for the 3D printed parts of the microscope. These components were 3D printed using a Creality Ender 3 S1. Modified 3d printed parts from the original OFM design were modeled with Autodesk Fusion 360

using the free license for academic institutions. 3D models were exported as StereoLithography (.stl) files and used to generate G-code in Cura 5.1.0 (Ultimaker).

Due to different printing conditions and parameters from previous studies, optimization to local conditions in the Philippines was needed [15]. This optimization allowed our team to correct unwanted outputs like warping, deformations, and non-adhesion of layers of the 3d printed components. The optimized 3d printing temperature for the open-source microscope were 210˚C and 60˚C for nozzle temperature and heat bed temperature, respectively.

All printed parts were printed in a standard layer height of 0.2 mm with 20% to 30% infill density in a cubic infill pattern. The wall thickness was set to 0.8 mm with a line count of 2. Printing speed was set to 50 mm/s for the whole experiment. The printed materials have a brim build plate adhesion type, except for the main body, and have no supports. Post-processing of the 3D printed parts was done to smoothen rigid surfaces and to remove brims/stringing.

Electronic, optical, and mechanical components of the microscope were purchased in the Philippines from various suppliers. These bought materials are various bolts and nuts, wire cables, three (3) 28BYJ-48 stepper motor controllers, 5mm LED lights, various resistors, three (3) RMS objective lens (10x, 40x, and 100x), achromatic lens, condenser lens, Raspberry Pi 4 model B+ (Rpi4B+), Raspberry Pi camera v2 (with Sony IMX219 8 MP), power supply, Arduino nano, and a soldering kit. Detailed descriptions of these components are presented in Tables 1 and 2.

The components were assembled starting from tightening the main body with the Viton O-ring with its gears and stepper motors. Afterwards, the optics module was assembled. Mounting the optics and the microscope was done before assembling the illumination component. Lastly, wiring for illumination, stepper motor and Rpi were done prior to software installation.

The software component used for the microscope was the Raspbian-Openflexure installed in the Raspberry Pi 4 model B. Microcontroller code for the stepper motor was installed in the Arduino Nano. The image collection commands were sent to the Rpi 4B containing a modified python-based color correction code.

All deidentified pathology slide samples used in this study were provided by the Ateneo School of Medicine and Public Health and are regularly used in pathology teaching modules. These samples had a known interpretation and were labelled with the concurrent pathology affecting the tissue and/or sample. The samples were observed on OpenFlexure Connect 4.0.1. The three samples used are labeled as rheumatic heart disease (RHD), tuberculosis (TB) in soft tissue, and ascariasis. Comparator sample pathological images were obtained using a mobile phone camera which took digital pictures from the same samples loaded in a LED-based light microscopes as reference.

## Image resolution calculation

Image resolution was calculated by first obtaining clear images at 40x and 100x of a calibration slide with line precision of .01 mm to .1mm using the OFM and a conventional LED light microscope which was used as the comparator.

Afterwards, ImageJ software (US NIH) was used to determine the image resolution automatically. This was done by measuring the pixel aspect ratio and the distance between each line subdivision of the calibration slide. The known distance between two lines, and the measurement obtained using the software, results in a calculated resolution in pixels/mm. It should be noted that the images obtained for LED light microscope and OFM were iPhone 12 camera (12 MP) and Raspberry Pi Camera v2 (8 MP), respectively.

**Table 1. Complete list of parts, components and items used in fabricating and using the OFM.**

| Name of Component | Description | Source | Remarks |
|---|---|---|---|
| Laptop | Standard windows 11 laptop, 256gb HD, 512gb SSD, 32gb RAM | Purchased off the shelf in Metro Manila, Philippines | The laptop was primarily used as the monitor and graphical user interface to control the OFM and visualize samples. Other monitors with HDMI inputs can be used as alternatives. The laptop also has inbuilt mouse and keyboard which is required to operate the OFM software. |
| Digital Monitor | Any computer monitor or television with High-definition multimedia interface (HDMI) inputs could be used to visualize samples from the OFM | (optional) locally available in any Philippine appliance store | Could be used in lieu of a laptop. |
| Keyboard | Any USB keyboard for controlling movement of the stage, saving and annotating images, general control for OFM | (optional) locally available in any Philippine appliance or computer store | Could be used in lieu of a laptop. |
| Optical Mouse | Any USB optical mouse for interacting with the user interface of the OFM software | (optional) locally available in any Philippine appliance or computer store | Could be used in lieu of a laptop. |
| Raspberry Pi 4B+ (Rpi4B+) | Microcomputer with minimum specifications of 2gb storage and RAM for installing OFM software. | Purchased online, imported | The Rpi is used as the onboard computer of the OFM, where the pre-built Raspbian-Openflexure (the operating system) is installed in an SD card. In addition, this OS contains all the necessary software, the Openflexure server and Openflexure Connect, to operate each OFM unit locally and over the internet. Moreover, sample images could be saved. The Rpi is also responsible for general input and output connections including internet Wi-Fi, Bluetooth, LAN |
| Power Supply | To power the Rpi4B+ and stepper motors | Purchased online, imported | Micro USB B for Rpi 3B and USB C for Rpi 4B. The voltage and ampage of the stepper motor adapter must be considered. |
| Stepper motors and Connection board (3x per unit) | 28BYJ-48, 5V, 4096 steps per revolution | Purchased online, imported | Moves and controls the stage of the microscope |
| 3d Printer and PLA filament | Ender S1 Pro and eSuns PLA filament | Purchased online, imported | Assembly and installation of printer can be done using the accompanying manual. This equipment was used to print the plastic components of the OFM |
| RMS threaded Objective lens | 45 mm parfocal length, finite conjugate, 160 tube length, plan corrected, (10x, 40x, 100x) | Purchased online, imported | Magnification lenses to visualize sample slides |
| Achromatic lens | glass, focal length 12.7 mm, 50 mm focal length | Purchased online, imported | Cemented achromatic doublet lens with antireflection coating for visible wavelengths is recommended. Used to refract light without spectral color separation. |
| Condenser lens | Polymethyl methacrylate (PMMA), 5 mm focal length, 12.7 mm outer diameter | Purchased online, imported | Concentrates light for illumination |
| LED light | 5 mm, warm light, or white light. | Purchased online, imported | Provides primary light source for illumination |
| Bolts and Fasteners | M3 nut, M3x10 screws, M3x25mm hex bolt, M4x6mm screws, Viton O-ring | In-store and online purchase | Stabilizes the components and used to assemble all parts into a single unit |
| Arduino Nano | Atmega328 microcontroller (version 3.0), 45 x 18 mm | Purchased online, imported | Controls the stepper motors. It requires a separate power adapter |
| Wires, jacks, barrel jacks, and connectors | connectors | In-store purchase | For wirings |
| Raspberry Pi Camera | version 2.1, 8 MP Sony IMX 219 | Purchased online, imported | Serves as the 'eyepiece' to visualize slide samples and the main digital image sensor |
| OpenFlexure Software | Openflexure connect 4.1, Windows x86–64 | Downloaded via their website opeflexure.org and is free for use | Interface of the microscope for control, visualization, storage |
| MicroSD card | (up to 32gb storage ONLY) | In-store and online purchase | Contains the Raspbian operating system or the Rpi4B+ |

**Table 2. 3D printed parts of OFM and their functions.**

| Parts/components | Usage |
|---|---|
| Actuator assembly | Supports Viton O-ring installation without damaging the flexure joints |
| Cable tidies | Hides wires inside the OFM |
| Sample clip | Handles the sample slide |
| Small and large gears | Causes movement of the stage |
| Illumination dovetail | Holds the condenser lens and LED light |
| Illumination thumbscrew | Secures the dovetail in place |
| Optics Pi-camera RMS (f50d13) and cover | Holds the Pi-camera and condenser lens |
| Feet | Holds the Viton O-ring |
| Lens tool | Secures the lenses |
| Microscope stand | Contains the Raspberry Pi 3B/4B |
| Motor driver case | Carries the Arduino nano |

## Image comparison of pathologic teaching slides

Using the same pathology teaching slides, images were taken using the OFM and a mobile phone to take a slide image from an LED light microscope. The team then examined the resulting images from each microscope for the teaching slides. For the TB in soft tissues slide, we identified caseating necrosis. For Rheumatic heart disease, we identified possible 'Aschoff's nodules' and 'Aschoff bodies' which are considered pathognomonic features of the disease under the microscope. For the Ascariasis slide, we identified ascaris eggs. Once we have identified the probable features, these were cross-referenced with a pathology atlas and manual [16] to verify if our interpretation of the features per slide was correct.

## Recording of expenditures and calculation of OFM fabrication cost per unit

Costs we recorded reflect the materials consumed in fabricating a single unit of the OFM. The costs for each material are calculated by dividing the quantity used in the OFM over the total number of a minimum order quantity multiplied by the purchase price for certain items like screws, washers, and bolts. The cost for single items like the Rpi 4B+ or RMS threaded objective lenses are directly recorded from their purchase prices. The costs incurred for the 3d printed components were calculated by taking the weight of each part, summed, and then divided by the total filament weight of 1kg. The resulting weight percentage is then multiplied to the purchase price for 1kg of filament to arrive at the cost for the 3d printed parts. For most of the purchases, delivery and shipping costs are already included and are reflected in the final purchase prices.

Cost comparison of OFM with other microscopes were done by scouring local online marketplaces and recording their listed price. These were then summed up with their features included to compare it with our fabricated OFM. All cost recorded and calculations were done using Microsoft excel.

## Modifications on the 3D printed microscope

The open-source, 3D printed microscope was successfully fabricated using the third iteration of the 7[th] version of the OFM design (Figs 1 and 2).

The printing temperature was optimized to 210˚C to achieve an intact and firm layer adhesion. Introduction of brims in all components fully minimized warping. Such a phenomenon must be controlled since the Philippines has a humid season and it can affect the mechanical

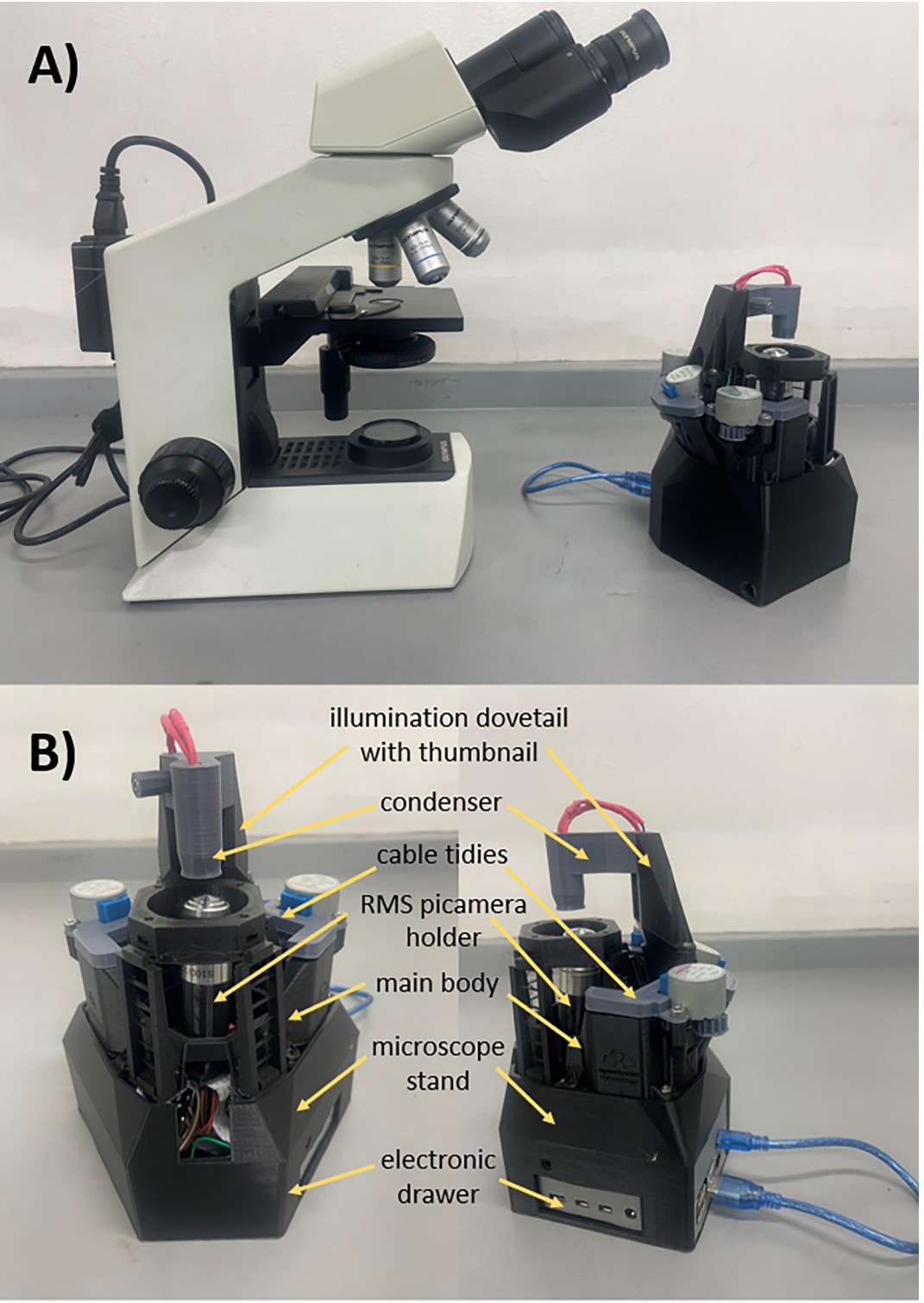

**Fig 2.** (A) Comparison of light microscope and OFM. B) Labeled parts of OFM.

motion of the flexure stage. Proper build adhesion is achieved when the heat bed temperature is 60˚C.

The main body was studied with eight (8) print trials to observe the structural integrity of the flexing joint part. One of these iterations was a custom printed snap ring fitting (Fig 1B). Even though the condenser lens purchased has the appropriate specifications, it could not fit properly to the condenser arm necessitating the fitting. The condenser lens creates a focused illumination cone to the slide sample.

Another iteration done was the power supply for the stepper motor. Based on the wiring system of the 3D printed microscope (Fig 1C), a problem regarding power distribution on the stepper motor was observed when the power supply used was 5.0 V with 1.0 A. No motion was observed at these specifications. However, motor motion was observed when the voltage was raised to 6.5 V. It was also observed that there was a delayed response (~ 1 min) when these parameters were used. To compensate for this observation, a voltage of 5.0 V with current of 3.0 A was implemented. Notable stability for motor control was observed using these parameters.

The camera is directly connected to the Raspberry Pi 4B through a ribbon cable. The infill density used was between 20% to 30% in a cubic pattern for all parts, except for the main body. The main body component was set to only contain an infill density of 20% to maintain the flexibility of the plastic joint without affecting its good mechanical stability [17–21]. However, the flexible joints of the main body are susceptible to deprecation at lower infill density. It was observed especially when the flexible joint exceeds its maximum translation distance from the objective lens. Increasing the infill density reduces such deprecation but restrains the translation mechanism. Continuous study of structural strength is still in progress using infill patterns. LED light color can be modified to replicate some light microscopes. Warm white illumination can be used to emulate halogen-based light microscopes and clear white light to emulate the LED-based microscopes.

### Ethics statement

This study was reviewed and approved by the Ateneo de Manila University research ethics committee (Study number: SMPH LFM 2022). No human subjects were involved in the building and initial image examination of the OFM prototype except for deidentified fixed pathology teaching slides.

## Results and discussion

### Image resolution of the OFM

Using ImageJ software, the image resolution was calculated for the OFM and LED microscope mobile phone captured images at 40x and 100x with the results displayed in Table 3.

**Table 3.  Image resolution at 40X and 100x for OFM and LED microscope.**

|  | OFM | LED light microscope (captured using mobile phone) |
|---|---|---|
| **40x Objective Magnification** | 6820 pixels/mm | 6044 pixels/mm |
| **100x Objective Magnification** | 18950 pixels/mm | 16480 pixels/mm |

The image resolution ratio at 40x between the OFM and LED microscope is ~1.13. At 100x this ratio is ~1.15. These results show that the OFM has comparable image resolution compared to the images from the LED light microscope captured by a mobile phone camera.

## Imaging of pathological samples from the 3D printed microscope

The camera setting is very important in obtaining optimal images. The camera sensor IMX219 is the most appropriate for the OFM since it is compatible with the Raspbian-openflexure operating system. We attempted to use a lower cost OV5647 5 MP camera sensor and were able to obtain images. However, appropriate color correction cannot be attained even with lens shading correction prescribed by Bowman [20]. Further study and modifications in making this camera sensor compatible with the software is in progress.

The 3D printed microscope created from available materials in the Philippines works well in scanning pathological sample slides. High-resolution images of RHD, TB in soft tissues, and Ascariasis pathological slides at 40x and 100x (without oil) can be obtained (Fig 3A–3L). The images obtained using the OFM (Fig 3A, 3B, 3E, 3F, 3I and 3J) showed similar colors from the reference (Fig 3C, 3D, 3G, 3H, 3K and 3L).

We examined the resulting images and identified the features per slide that we were able to, as referenced with a pathology manual since the teaching slides had known interpretations. The RHD slides were difficult to interpret, but there appeared to be the presence of Aschoff nodules as evidenced by multinucleated cells (Fig 3B). The 40x magnification RHD image was more difficult to interpret due to the age of the slide and the color correction of the OFM. The TB in soft tissues showed a granuloma which is indicative of caseous necrosis (Fig 3E and 3F). The sample ascariasis slide, without any staining had clearly visible ascaris eggs but with no brown color due to the sample having faded with time (Fig 3I and 3J). These observations are promising, but we intend to show these sample images to a pathologist to verify our findings and to do more tests to improve the focus of our captured images.

## Accessibility and affordability of components and 3D printed parts

The components of the microscope were sourced from local suppliers in the Philippines, whether online or in-store purchase. The 3d printed plastic microscope parts took 52 hours to be printed and used a total filament weight of ~ 350 g. The assembly time of the microscope requires approximately an hour. The assembled microscope weighs 565 g with a total cost of ~310 $ US per unit (~Php 17,051.00 using 55PHP to 1$ US exchange rates and rounded up using December 28, 2022 rates, see Table 4 for breakdown) without labor and electricity consumption costs.

Previous builds of the OFM quoted the high-optics version at $200 in parts [17]. Our own version cost a total of ~$310. The difference may be because we accounted for all components and some accessories, inflation, importation fees, alongside increases in the landed costs of these items within the Philippines. From Table 4, the largest contributors to the total cost are the 1) Rpi 4B, 2) Rpi camera vr.2, and 3) the achromatic lens. These three components contribute 70% to the total cost of the OFM unit. These three items and the objective lenses were some of the most difficult to procure items for the OFM and may reflect persistent supply chain issues [22, 23]. Table 5 contains a non-exhaustive list of commercial microscopes, their features, weight, and prices alongside our OFM unit.

From this small selection of commercial units, our OFM has the lowest cost amongst microscopes with digital cameras. In combination with its advanced features and connectivity, this gives the OFM more utility for its cost compared to the other noted microscopes in the table.

About 62% of the microscope's weight came from the printed parts. Compared to the usual compound microscope, it weighs ten times lighter. This microscope is portable and can be readily deployed in facilities with existing power and space to house microscopes.

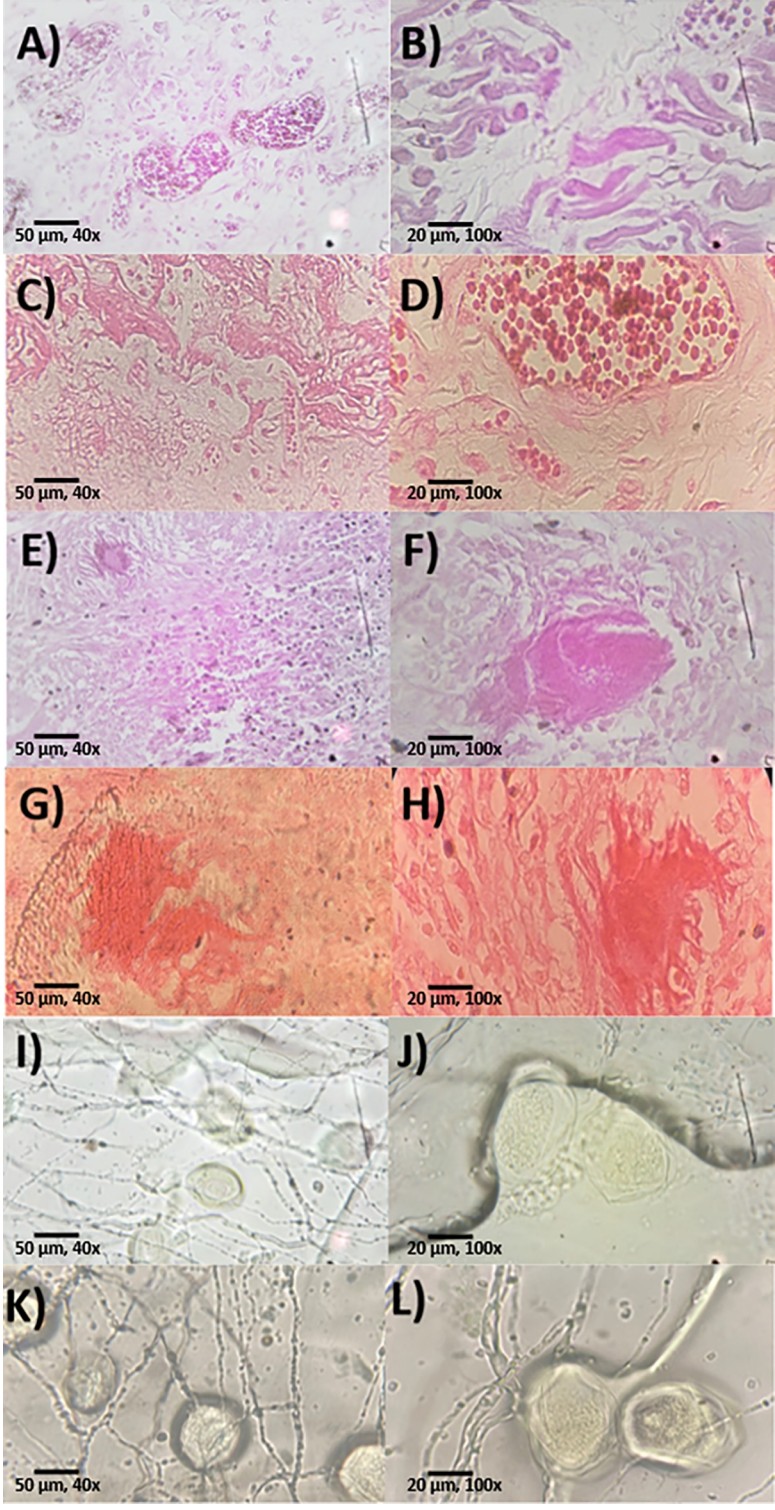

**Fig 3. JPEG file images of pathology teaching slide samples.** Images A, B, C and D are samples of rheumatic heart disease. A and B were taken using the OFM, while C and D were taken using a mobile phone from a LED light microscope. Images E, F, G and H are tuberculosis in soft tissues with E and F being captured using the OFM, while G and H are from the LED microscope. Images I, J, K, and L are ascariasis samples with samples I and J being taken using the OFM. Samples labelled as K and L were samples observed using LED-based light microscope. Samples C, G, and K at 40x and samples D, H, and AL at 100x magnification were used as reference to observe the appropriate color expected from the camera sensor of the OFM under a white light LED.

**Table 4. Detailed breakdown of the costs of the OFM components.**

| Component used | Quantity Used | Cost per quantity in US Dollars $ | Cost in Philippine Peso ₱ | Cumulative share in total cost |
|---|---|---|---|---|
| Raspberry Pi 4B, | 1 | $ 144.00 | ₱ 7,920.00 | 46.46% |
| Raspberry Pi Camera vr.2 | 1 | $ 42.65 | ₱ 2,346.00 | 13.76% |
| Achromatic Lens | 1 | $ 30.49 | ₱ 1,677.20 | 9.84% |
| Arduino Nano | 1 | $ 15.25 | ₱ 839.00 | 4.92% |
| MicroSD card (32gb) | 1 | $ 14.54 | ₱ 800.00 | 4.69% |
| Power Supply 5V | 2 | $ 12.50 | ₱ 688.00 | 4.03% |
| RMS threaded microscope objective 100x | 1 | $ 11.85 | ₱ 652.00 | 3.82% |
| M3 Nut | 14 | $ 5.60 | ₱ 308.00 | 1.81% |
| RMS threaded microscope objective 40x | 1 | $ 5.60 | ₱ 308.00 | 1.81% |
| RMS threaded microscope objective 10x | 1 | $ 5.20 | ₱ 286.00 | 1.68% |
| Stepper motor with integrated board | 3 | $ 5.15 | ₱ 283.50 | 1.66% |
| Polylactic acid (PLA) filament | ~350g | $ 4.80 | ₱ 266.00 | 1.55% |
| M3x10 cap head screws | 10 | $ 2.20 | ₱ 120.00 | 0.71% |
| HDMI Cable | 1 | $ 1.81 | ₱ 100.00 | 0.58% |
| Ethernet Cable | 1 | $ 1.35 | ₱ 74.50 | 0.44% |
| M4x6 bottom head screws | 6 | $ 1.29 | ₱ 71.00 | 0.42% |
| Jumper Cables | 14 | $ 1.27 | ₱ 70.00 | 0.41% |
| Solder wire and PCB board | 1 | $ 0.81 | ₱ 45.00 | 0.26% |
| M3 Stainless steel washers | 8 | $ 0.79 | ₱ 43.20 | 0.25% |
| Condenser Lens | 1 | $ 0.78 | ₱ 43.00 | 0.25% |
| Viton O-ring | 3 | $ 0.74 | ₱ 40.50 | 0.24% |
| M3 Brass nuts | 3 | $ 0.64 | ₱ 35.34 | 0.21% |
| M3x25 cap head screws | 4 | $ 0.45 | ₱ 25.00 | 0.15% |
| M3x8 cap head screws | 1 | $ 0.10 | ₱ 5.00 | 0.03% |
| Heat Sink | 1 | $ 0.04 | ₱ 2.00 | 0.01% |
| Resistors 150 Ohms | 1 | $ 0.03 | ₱ 1.53 | 0.01% |
| 5mm LED | 1 | $ 0.02 | ₱ 1.00 | 0.01% |
| Total Cost | | $ 309.95 | ₱ 17,050.77 | 100.00% |

## Supply chain implications

As shown above, we have demonstrated that 3D printing and using locally accessible and affordable supplies can be utilized to build essential diagnostic equipment. Our motivation for this approach is to develop resiliency and the capacity to sustain and maintain core equipment used in clinical and public health services in the Philippines which is almost completely dependent on importation for its medical equipment and supply needs [30]. In addition, the COVID-19 pandemic revealed how fragile the supply chain for essential medical devices and supplies were, especially for developing countries like the Philippines which has minimal ability to manufacture the technology needed to respond to health shocks and disruptions [13]. A second problem is the difficulty with procuring high-tech equipment and supplies in the Philippines. Anecdotes have highlighted that the process is complex, difficult, and time-consuming [31] which result in long turn-over times and bidding failures. This may also be the case for medical equipment and supplies, especially for health facilities in rural and underserved areas. While these problems might not be directly solved by building lower cost technology in-country, local procurement policies typically favor lower cost items and suppliers [32].

Additive manufacturing and open-source approaches enable distributed manufacturing (DM) [17] and point-of-care manufacturing (POCM) [33]. Having design, manufacturing, and

**Table 5. Sample of models and prices of laboratory microscopes in Philippine marketplaces.**

| Microscope Brand | Features | Weight (g) | Price $ (₱) |
|---|---|---|---|
| OpenFlexure microscope | One camera with objective lens (switchable 10x, 40x, 100x)<br>LED illumination<br>Automated stage (x-, y-, and z-axis)<br>Tile scanning<br>z-stacking<br>slide scan | ~500 | $310<br>(₱17,051) |
| Olympus CX- 23 [6] | Binocular head, 360˚ rotatable<br>10x focusable eyepieces with large 20mm field of view<br>4x, 10x, 40x, 100x (oil) infinity plan objectives<br>LED transmitted illumination | 5900 | $1295<br>(₱ 74,250) |
| OM36 compound microscope [24] | Binocular microscope<br>4 Objective Lenses - 4x, 10x, 40xs, 100xS oil<br>Binocular or trinocular<br>Halogen or LED illumination<br>Full professional features<br>Included camera adapter | 1100 | $649<br>(₱ 35,695) |
| Meiji Techno TC-5100 [25] | Two LWD objectives 10x, 20x<br>SWH 10x eyepiece<br>Binocular or trinocular<br>30W halogen illumination<br>No camera | 10000 | $5750<br>(₱ 316,250) |
| AmScope 40X-2500X Infinity Kohler [26] | Optical System Infinite-conjugate<br>Head Trinocular, 30˚ incline, 360˚ rotatable.<br>10 MP Camera<br>4 Objective lenses– 4x, 10x, 40xS, 100xS oil<br>Halogen bulb | 5440 | $1837<br>(₱ 101,035) |
| GEEFUNTECH HD [27] | Objectives: 195 HD objective lens DIN 4X, 10X, 40X(S), 100X (S, Oil). Adjustable ocular diopter<br>2MP image sensor<br>Operating system: Windows XP, Vista, 7/8/10, Mac 10.5 | 1700 | $645<br>(₱ 35,500) |
| Kern OBS 101 [28] | Manual operation<br>Battery operated.<br>Height adjustable<br>4x magnification<br>No camera | 2500 | $262<br>(₱14,451) |
| Optika SFC-3AF [29] | Manual operation<br>Coarse focusing adjustment<br>3 objective lenses– 4x, 10x, 40x<br>No camera | 1300 | $87<br>(₱ 4, 769) |

maintenance capacities closer to the point of use allows rapid and contextualized responses to local needs. This enables a more rapid design change response to adapt a piece of technology to challenges in the field.

The emergence of desktop 3d printers that are affordable, available in the Philippine market, and easy to use, encouraged our team to purchase this equipment. These tools are not free, but the capital investment appears affordable at $372.73 (₱20,500 at ₱55 to $1 December 28, 2022, exchange rates). Considering that spare parts, support, and an active 3d printing community online and within our university is thriving, this appears to be a worthwhile investment to complement traditional manufacturing and in building more localized capacity.

For some regions, 3d printing became a large part of the COVID-19 response as it allowed hospitals nearer the point of care to rapidly design, manufacture, test, and deploy essential equipment. Examples of 3d printing being used to produce equipment include face shields, ear savers, masks, ventilator adapters, connectors, hands-free door openers, and other ports [34].

## Implications on access to diagnostic microscopy

At present, our OFM version appears to have a similar image quality to those obtained by light microscopes combined with mobile phone cameras as evidenced by our image resolution results in Table 3. This is a promising result considering that the per unit cost of fabricating one unit is at 310 $ US, which is four times cheaper than a brand-new laboratory grade compound light microscope, the CX Olympus series, typically used in local clinical microscopy. The unit price is also cheaper compared to other digital microscopes in the market with similar features as shown in Table 5.

Aside from the cost, the ability to 3D print parts and procure readily available components makes supporting our prototype in the field a viable prospect. This flexibility and local capacity to fabricate parts for more accessible, affordable, and available repairs could potentially reduce downtime and unavailability of diagnostic microscopy services in public health facilities, especially those at the primary care level.

For its weight and size, our OFM prototype contains powerful features. Having a micro-computer integrated into the unit allows our microscope to connect and interface with different technologies including mobile phones, desktop computers, laptops, and internet-enabled digital displays. It can do this over a direct cable connection, a local area network, Bluetooth, or by connecting to the internet through wi-fi. The existing OFM software interface also allows for precise, simple slide scanning (x-y planes), z-scanning and stacking, and viewing in real time and storing digital captured images of the slides within the on-board computer. The captured images can be as small as 3.5 Megabytes per image. In a microSD card with 8 gigabytes of storage it would be possible to store ~2,200 images, assuming each image was 3.5MB. In instances of needing more space, a larger capacity microSD card can be bought and used with the OFM or an external hard drive can also be attached for easy transfer of images locally. If the unit is connected to the internet these digital slides can be easily sent to a secure central repository in another computer or online in a secure dedicated internet server storage. An under-appreciated advantage of the OFM over conventional microscopes is the ability to obtain images up to 1000x magnification without needing oil whether using an 100x oil immersion or 100x dry objective. This reduction of oil usage can help reduce operational costs per test, making it easier to maintain the unit without sacrificing magnification.

Another promising use case of the OFM is in telepathology. Primary care facilities can implement hub and spokes service delivery models [23] for microscopy wherein pathology experts or other trained personnel can analyze digital images collected from field sites. Considering acute shortages of health personnel [24], telepathology may help address some of the human resource gap and offer improved access to expertise and support to frontline facilities and personnel that may not be available in their localities using either real-time videoconferencing or store and forward approaches. The prototype unit itself is portable given the total weight and size, allowing easy transport between different types of health facilities.

Evidence suggests that telemedicine, which may also apply to telepathology, could be cost effective for access to specialist services and may lead to improved outcomes. A national telepathology project evaluated in China showed cost savings of ~$50 per patient and substantial savings of ~$300,000 for the centralized system. Furthermore this same study demonstrated improvement in the turnaround time to releasing pathology reports year-on-year from 29.36 hours to 9.75 hours from 2016 to 2019 [35]. Another study in Bangladesh showed that telepathology provides access to populations to rural areas. The study also concludes that telepathology is 58% more cost effective compared to conventional pathology services. The telepathology services had a total cost of $9.9 vs $23.8 with regular pathology services per diagnosis made [36].

In the Philippines, while no study has been done on telepathology, the population has come to appreciate the role of telemedicine for availing healthcare services and is broadly acceptable as a convenient option [37]. Further research should be done to investigate the cost effectivity of telemedicine and telepathology at the local level [38, 39].

It is unknown how many among the 2,593 rural health units (RHUs) or government primary care centers in the Philippines face difficulty in fulfilling the licensing requirements due to deficiencies in their clinical diagnostic services involving microscopy. However, projections by the department of health (DOH) show that 6,862 new primary care facilities would be needed to meet current and future healthcare needs [40]. In this scenario, it is possible that our prototype OFM can be deployed in these primary care facilities as the primary or adjunct equipment to enable access to diagnostic microscopy through telepathology.

Another alternative to conventional light microscopy that has been attractive for global health applications is mobile phone microscopy [41]. The ubiquity of mobile phones and their computing power today can match the specifications of desktop computers and laptops. Multiple studies have evaluated mobile phone microscopy for different applications including tuberculosis, soil-transmitted helminthiasis, schistosomiasis, and cervical cytology. In these evaluations, larger organisms such as helminths and schistosomiasis eggs are identifiable because of their larger size. However, the performance of these mobile phone microscopes varies. For instance, the diagnostic accuracy of the Foldscope mobile phone attachment for cervical cytology on de-identified samples reached 80% with a sensitivity and specificity of 85 and 90% for the high-grade squamous intraepithelial lesion (HSIL) category, 80 and 83.3% for low-grade squamous intraepithelial lesion (LSIL), and 70 and 96.7% for normal slides [42]. In another study evaluating mobile phones to detect *S. Mansoni* and *T. trichiura*, the sensitivity and specificity of the mobile phone setup was 68.2% and 71% respectively [43]. A third study comparing a handheld microscope (Newton Nm1) with the Cellscope mobile phone microscope station found that the latter had a sensitivity and specificity of 50% and 99.5% for *S. Mansoni*, and 35.6% and 100% for *S. haematobium* [44].

While these evaluations are promising, the uptake of mobile phone microscopy into regular clinical or public health services, especially in developing countries, has been poor. Reasons for this may include significant training needs to utilize these mobile phone microscopes, complex sample preparation procedures that require other equipment, regulatory hurdles and quality assurance, the need for viable manufacturing and business models, and regular maintenance [45].

As an open-source design, there is the opportunity to modify elements of the microscope to help localize and adapt for specific use situations in the Philippines [46]. Our team intends to collaborate with pathologists, primary care physicians, medical students, biologists, and medical technologists to further refine and identify modifications needed to aid in localizing and adapting our current prototype by examining its usability and features. For instance, our team is exploring more expansive use of controllers with the OFM software because this is closer to the workflow used by physicians and medical technologists in the field. This is done by manually assigning keys or buttons of either the keyboard or another controller to control the z-steps. This allows the users to manually move and focus a slide sample. Another attractive feature of our prototype being open source is how the OFM can be utilized as a platform for other visual diagnostics like fluorescence microscopy [8].

## Limitations of the OFM

Based on our initial assessment of the OFM, it does have several limitations and challenges that prevents its implementation to health facilities at this time. Especially when being

intended for deployment in a range of diverse contexts within the Philippine healthcare system. One consideration is its throughput in scanning entire slides. The OFM has precise stage movement but at the cost of rapid scanning and movement of sample images while being viewed [7]. Depending on the user-programmed scanning parameters, and slide sample size and depth, capturing a sub-section of a slide can take between 5 to 30 minutes or longer. Another limitation of the OFM is its inability to rapidly change magnification manually. This can be overcome by scanning and collecting digital images of different areas of a slide sample and stitching them together using software [17]. We are currently investigating possible design changes to allow for more rapid changes of magnification that closely align with existing microscopy practices. Related to this physical limitation, the OFM has only limited slide scanning features and cannot replicate whole slide scanning, where the area of the entire slide can be scanned and digitized in minutes [7].

Compared to conventional microscopy, the OFM does not have existing training or workflow protocols for specific clinical use cases or public health programs. In this context, further design work and training is needed to adequately implement this technology in the field and to adapt existing workflows to enable the potential of the OFM unit.

An important consideration for more rural and underdeveloped areas in the Philippines are the power requirements of the OFM. It does need electric power to operate, and for areas with intermittent or absent electricity this would lead to service disruptions. The OFM also requires accessories for operations. The unit needs a monitor, keyboard, mouse, and a computer or laptop to enable viewing of slides in real-time or asynchronously. To use telepathology for OFM, there also needs to be internet access within public healthcare facilities and this is not universally available especially for rural areas [47]. The advantages and disadvantages of the current configuration of our OFM are listed in Table 6.

The identified disadvantages of them OFM are not insurmountable from a design perspective and future iterations can be focused on improving the slide move speed, magnification changes and develop the necessary clinical protocols to utilize the OFM in current practice within Philippine primary care facilities. However, these limitations can dissuade stakeholders

**Table 6. Advantages and disadvantages of OFM vs. Conventional LED microscopes.**

| Unit | Advantages | Disadvantages |
|---|---|---|
| OpenFlexure | • Lightweight<br>• Wi-fi, Bluetooth, local area network adapter<br>• Partial slide scanning and z-stacking<br>• Automated focus features<br>• Digital slide capture and storage<br>• Readily available parts and relatively easy to repair 3d printed parts<br>• Precise stage movement<br>• Image resolution is slightly superior to conventional microscope images captured using a mobile phone<br>• For a digital microscope is affordable at $310<br>• Does not need oil to reach 1000x magnification even when using a 100x oil-immersion objective. Software correction enabled adequate image focusing regardless of whether the objective is 100x oil immersion or dry | • Slow slide movement speed<br>• Cannot rapidly change magnification manually<br>• Requires power and other accessories to operate<br>• Does not yet have clinical workflow or protocols for specific use cases |
| Conventional light microscopes | • Durable and when properly maintained has a significant lifespan of use<br>• Basic Units are readily available and some do not need power to be used<br>• Standard laboratory equipment for basic diagnostics<br>• Rapid change in magnification to zoom in and out of sample images<br>• Most healthcare workers trained in its use | • Microscopes with digital capture features can be expensive (see Table 5.)<br>• Does not have internet capabilities<br>• Slide scanning and stacking not available<br>• Maintenance can be difficult<br>• Requires storage of physical slides<br>• Difficult to transport<br>• Maintenance and replacement can be difficult |

in the local health system from obtaining and implementing this OFM unit. Further studies would need to investigate whether design changes, and other factors may lower the barrier to adoption. Implementing co-design principles with end-users to adapt and make the OFM fit-for-purpose especially for frontline healthcare workers and facilities in the public sector may also increase its potential to be adopted and implemented by health facilities in the Philippines [48].

## Conclusion, recommendations, and future directions

The image resolution of the OFM compared to those of images captured by mobile phones from an LED light microscope is slightly higher at 13% at 40x and 15% at 100x. We had attempted to use even lower cost cameras for the OFM unit but had persistent color correction issues until we reverted to the Raspberry Pi vr2 camera. We intend to pursue further optimization to improve lighting and color to enhance the focus and quality of the captured images using our OFM unit.

The unit we built was able to capture images up to 1000x without using oil for RHD, TB in soft tissues and ascariasis. When compared to images at the same magnification from a conventional microscope captured by mobile phone, similar colors and lighting were observed. In these teaching slide samples, we were able to identify Aschoff nodules, caseous necrosis and ascaris eggs. We will seek the expertise of a pathologist to validate our interpretation while aiming to improve the focus of the captured images in future work.

The fully assembled OFM unit weighed a total of 565g with 62% of this comprised of the 3d printed plastic parts. The total print time for all plastic components totaled ~52 hours with an additional 1 hour of assembly for both the electronics and plastic components.

Our study shows that with some investments in equipment and utilizing accessible, available, and affordable 3d printers and CAD design tools, we were able to fabricate and modify an open-source high-resolution digital microscope. It comes with the ability to capture and store digital slide images, scan slides digitally in real-time, connect to the internet, Bluetooth, and LAN, and have precise stage control. All these features cost ~$310 while being lightweight and portable with its own software available and free to use with the unit.

In the context of expanding access to essential diagnostic microscopy, the OFM can be used to help augment primary care facility microscopy services. Being able to produce these units locally can build capacity to help maintain, upgrade, or expand the functionalities of the OFM to meet existing and emerging health needs. This same capacity can also help mitigate the effects of global supply chain shocks [13]. Furthermore, local production can also be beneficial in accelerating procurement of high-tech equipment by health facilities.

The features of the OFM that enable digitalization include the ability to directly capture an image to a microcomputer, Internet wi-fi, Bluetooth or local area network connections, partial slide scanning, z-stacking, precise stage movement and control, lightweight, portable, and easily repairable and modifiable, does not need oil for 100X oil immersion objective magnification, and its cost of $310. However, while these features are powerful, the unit at present may not be completely attractive to public health facilities because of the following limitations. First are its power and accessory requirements. Second is its scanning throughput. Third is the lack of protocols and workflows for specific use cases. We intend to explore and develop protocols for using the OFM with procedures like gram-staining, pap-smears, TB microscopy including giemsa-staining and fluorescence techniques, urinalysis, blood-cell counting, fecalysis and Kato-Katz stool examination techniques. Fourth, is that training, education, and user-guide documentation are needed to develop the necessary expertise to use the OFM for any diagnostic procedures, including those we listed above, and service standards specified by the DOH.

Lastly, to maximize its chances of being adopted and implemented by health facilities, additional design work and features development are needed to improve the unit for local use contexts especially in under resourced areas. This can include integration of the unit with electronic health records and telemedicine systems already being implemented in different health facilities.

Considering the present limitations of the OFM, future work in advancing our prototype to enable its use by health facilities for diagnostic microscopy services includes undertaking a usability assessment and subsequent iteration with pathologists, primary care physicians, recent medical graduates, undergraduate medical students, medical technologists, and biologists for a variety of use case scenarios. Improvements in the sharpness, focus, and scanning protocols for different use cases would also be explored with the target end-users. Moreover, a priority for further research is to generate evidence on the diagnostic performance of our OFM prototype, and to adapt operational protocols using the OFM specifically for those service standards stated in the Philippine Department of Health Administrative Order 2021–0037.

## Acknowledgments

We would like to thank the ASMPH center for research and innovation (ACRI) for their support in this project, especially Ayedee Ace M. Domingo, Innovation Flagship Head. We would also like to acknowledge Ricardo Jose Guerrero, Research Fellow in the Ateneo Research Institute for Science and Engineering (ARISE) for his input and guidance in the development of our OFM prototype.

## Author Contributions

**Conceptualization:** Jeremie E. De Guzman.

**Data curation:** Mark Kristan Espejo Cabello.

**Formal analysis:** Mark Kristan Espejo Cabello.

**Funding acquisition:** Jeremie E. De Guzman.

**Investigation:** Mark Kristan Espejo Cabello.

**Methodology:** Mark Kristan Espejo Cabello, Jeremie E. De Guzman.

**Project administration:** Mark Kristan Espejo Cabello, Jeremie E. De Guzman.

**Resources:** Mark Kristan Espejo Cabello.

**Software:** Mark Kristan Espejo Cabello.

**Supervision:** Jeremie E. De Guzman.

**Validation:** Mark Kristan Espejo Cabello.

**Visualization:** Mark Kristan Espejo Cabello.

**Writing – original draft:** Mark Kristan Espejo Cabello, Jeremie E. De Guzman.

**Writing – review & editing:** Mark Kristan Espejo Cabello, Jeremie E. De Guzman.

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
