## [Decision Letter · Decision Letter 0]

7 Mar 2023

PGPH-D-22-02112

Utilization of accessible resources in the fabrication of affordable, portable, high-resolution 3D printed digital microscope for Philippine diagnostic applications

Dear Dr. De Guzman,

Thank you for submitting your manuscript to PLOS Global Public Health. After careful consideration, we feel that it has merit but does not fully meet PLOS Global Public Health’s publication criteria as it currently stands. Therefore, we invite you to submit a revised version of the manuscript that addresses the points raised during the review process.

We look forward to receiving your revised manuscript.

Kind regards,

Cesar Ugarte-Gil, MD, MSc, PhD

Academic Editor

Journal Requirements:

1. Please send a completed 'Competing Interests' statement, including any COIs declared by your co-authors. If you have no competing interests to declare, please state "The authors have declared that no competing interests exist". Otherwise please declare all competing interests beginning with the statement "I have read the journal's policy and the authors of this manuscript have the following competing interests:"

Additional Editor Comments (if provided):

The manuscript is well written, however there are some comments and doubts identify by the reviewers before to be accepted by the journal.

Reviewers' comments:

Reviewer's Responses to Questions

**Comments to the Author**

1. Does this manuscript meet PLOS Global Public Health’s publication criteria? Is the manuscript technically sound, and do the data support the conclusions? The manuscript must describe methodologically and ethically rigorous research with conclusions that are appropriately drawn based on the data presented.

Reviewer #1: Partly

Reviewer #2: Yes

2. Has the statistical analysis been performed appropriately and rigorously?

Reviewer #1: I don't know

Reviewer #2: N/A

3. Have the authors made all data underlying the findings in their manuscript fully available (please refer to the Data Availability Statement at the start of the manuscript PDF file)?

Reviewer #1: Yes

Reviewer #2: Yes

4. Is the manuscript presented in an intelligible fashion and written in standard English?

Reviewer #1: Yes

Reviewer #2: Yes

5. Review Comments to the Author

Reviewer #1: Material and Methods:

Lines 84 – 91: If authors decided to mention the list of materials, they should share a full description of the list instead of listing few parts. For readers, it can be convenient to know which components were available in the Philippines and which were imported or replaced by others.

Lines 92, 93: Authors should explain why they used different CAD software. Authors should highlight Fusion 360 has only a 30-days trial for free use and Sketchup Pro is a paid software. These lines are not consistent with the abstract and conclusions.

Line 95, 96: Due to this journal is not an engineering journal, authors should be explained more about the files, how they were generated, and in which stages they were relevant for manufacturing.

Lines 124, 125: What does “temperature was optimized” mean? Were the 3D printer or extrusion system modified for this purpose? 3D Printer temperature can be set easily using the software or hardware interface.

Results and Discussion:

Fig 1: How was the Open Flexure Microscope (OFM) used? A keyboard, mouse, and a screen were connected? How were the Arduino and Raspberry Pi connected each other? Please, share more images from the setup and it is recommend a short video to demonstrate slide scanning capability of the OFM.

Lines 124, 125: What does “temperature was optimized” mean? Were the 3D printer or extrusion system modified for this purpose? 3D Printer temperature can be set easily using the software or hardware interface.

Lines 127-129: What does “trials” mean? To print several times or perform mechanical tests?

Fig 2: All the images should have a scale reference for showing the OFM optics capabilities.

Lines 188, 189: Raspberry Pi storing capacity does not depend on computer capacity, but it depends on SD cards or external storages.

Lines 190, 191: Why were not the lens of the camera or the web cam version of the OFM used in this study?

Line 195, Table 1: It could be useful see a detailed list of costs per components.

Line 224, 225: In this study, it was used only an open-source design but the manufacturing tools such as CAD software and 3D printer were proprietary technologies.

Lines 231, 232: This is speculation. No tests or comparison were documented in this article. Only reports from end-users were shared.

Lines 233-235: It could be useful to have model references and prices from the Philippine market.

Lines 305, 306: Cite reference for this sentence.

Conclusions and future directions

Lines 325-327: How was tested these capabilities? Only visual examination? end-users were experts?

Reviewer #2: The Manuscript by Jeremie Eraña De Guzman and Mark Kristan Espejo Cabello on Utilization of accessible resources in the fabrication of affordable, portable, high resolution 3D printed digital microscope for Philippine diagnostic applications is an interesting well written paper which tries to highlight on the utilization of readily accessible and affordable resources in fabrication of high resolution microscope that can be used in diagnosis. This is one issue that we cannot run away from since it is widely accepted that, although microscopy plays an integral role in diagnostics, the currently available techniques and equipment are quite expensive and out of reach to most developing and underdeveloped countries. As such, the authors saw the need to design and developed an easy to use, low-cost Openflexure microscope prototype that will go along way in addressing the diagnostic challenges in most health facilities in resource limited regions.

The authors guide readers through the process of utilization of accessible resources in the fabrication of affordable, portable, high resolution 3D printed digital microscope. What stands out is actually that the prototype Openflexure microscope actually had a adequate clarity in identification of cellular features related to pathological features associated with infections under investigations. The paper is very well written with some edits to be made.

Besides finding the study to be a bit lacking in comparing and contrasting the findings of the OFM to other conventional microscopy, I have highlighted a few edits and suggestions for the authors to consider, as discussed in details in the attached document.

6. PLOS authors have the option to publish the peer review history of their article (what does this mean?). If published, this will include your full peer review and any attached files.

**Do you want your identity to be public for this peer review?** For information about this choice, including consent withdrawal, please see our Privacy Policy.

Reviewer #1: **Yes: **Pierre Padilla Huamantinco

Reviewer #2: No

---

## [Decision Letter · Decision Letter 1]

10 Jul 2023

PGPH-D-22-02112R1

Utilization of accessible resources in the fabrication of affordable, portable, high-resolution 3D printed digital microscope for Philippine diagnostic applications

Dear Dr. De Guzman,

Thank you for submitting your manuscript to PLOS Global Public Health. After careful consideration, we feel that it has merit but does not fully meet PLOS Global Public Health’s publication criteria as it currently stands. Therefore, we invite you to submit a revised version of the manuscript that addresses the points raised during the review process.

We look forward to receiving your revised manuscript.

Kind regards,

Cesar Ugarte-Gil, MD, MSc, PhD

Academic Editor

Journal Requirements:

Additional Editor Comments (if provided):

Reviewers' comments:

Reviewer's Responses to Questions

**Comments to the Author**

1. If the authors have adequately addressed your comments raised in a previous round of review and you feel that this manuscript is now acceptable for publication, you may indicate that here to bypass the “Comments to the Author” section, enter your conflict of interest statement in the “Confidential to Editor” section, and submit your "Accept" recommendation.

Reviewer #1: (No Response)

Reviewer #2: All comments have been addressed

2. Does this manuscript meet PLOS Global Public Health’s publication criteria? Is the manuscript technically sound, and do the data support the conclusions? The manuscript must describe methodologically and ethically rigorous research with conclusions that are appropriately drawn based on the data presented.

Reviewer #1: Yes

Reviewer #2: Partly

3. Has the statistical analysis been performed appropriately and rigorously?

Reviewer #1: I don't know

Reviewer #2: N/A

4. Have the authors made all data underlying the findings in their manuscript fully available (please refer to the Data Availability Statement at the start of the manuscript PDF file)?

Reviewer #1: Yes

Reviewer #2: Yes

5. Is the manuscript presented in an intelligible fashion and written in standard English?

Reviewer #1: Yes

Reviewer #2: Yes

6. Review Comments to the Author

Reviewer #1: General comments:

- To review the manuscript and correct the typos in the text and table content.

- To correct scale in images D and H from Figure 3.

- To explain why an inverted microscope (OFM) was built to compare it with an upright microscope (conventional light microscope). There is an upright microscope version of the OFM:

https://build.openflexure.org/openflexure-microscope/v7.0.0-alpha2/upright-microscope.html

https://openflexure.discourse.group/t/freshly-assembled-upright-microscope-v7-0-0-beta1/1151

Abstract:

- To mention "TB" means "tuberculosis" and not after.

- To verify if the OFM is a prototype or a final product. The term "prototype" suggests that this microscope is still far from being used in health facilities.

Materials and Method:

- To improve the description of the terms "OpenFlexure Operating System" and "OpenFlexure Connect" (GUI). In this section, the term "software" is used for both solutions but they have different purposes.

- To share the sensor resolution of the Raspberry Pi camera and smartphone.

- To share or update images where the ring for the condenser is used.

- If a smartphone is needed for digital microscopy in a real context, its price should be included.

Results and Discussion:

- This article points out the need for accessible instruments for microscopy; however, it is briefly discussed about implications of national regulation for medical devices. Can an improved prototype be used in health facilities? Please, provide more context or information about this topic.

Conclusions

- It cannot be concluded that the OFM is better than the mobile phones when only one model was used in this study. The quality and performance will depend on the camera sensor and lenses.

- Please, describe which kind of objective you used. For 100x, there are oil and dry objectives.

- Please, give more information about protocols to be adapted. Standard sample preparation protocols can be used with the OFM. User guides may be different due to the technology.

Reviewer #2: (No Response)

7. PLOS authors have the option to publish the peer review history of their article (what does this mean?). If published, this will include your full peer review and any attached files.

**Do you want your identity to be public for this peer review?** For information about this choice, including consent withdrawal, please see our Privacy Policy.

Reviewer #1: No

Reviewer #2: **Yes: **Collins Kipkorir Kebenei

---

## [Editor Report · Decision Letter 2]

4 Oct 2023

Utilization of accessible resources in the fabrication of an affordable, portable, high-resolution, 3D printed, digital microscope for Philippine diagnostic applications

PGPH-D-22-02112R2

Dear Dr De Guzman,

We are pleased to inform you that your manuscript 'Utilization of accessible resources in the fabrication of an affordable, portable, high-resolution, 3D printed, digital microscope for Philippine diagnostic applications' has been provisionally accepted for publication in PLOS Global Public Health.

Best regards,

Cesar Ugarte-Gil, MD, MSc, PhD

Academic Editor